# Laminin is the ECM niche for trophoblast stem cells

Daiji Kiyozumi[1], Itsuko Nakano[1], Ryoko Sato-Nishiuchi[1], Satoshi Tanaka[2], Kiyotoshi Sekiguchi[1]

The niche is a specialized microenvironment for tissue stem cells in vivo. It has long been emphasized that niche ECM molecules act on tissue stem cells to regulate their behavior, but the molecular entities of these interactions remain to be fully elucidated. Here, we report that laminin forms the in vivo ECM niche for trophoblast stem cells (TSCs), the tissue stem cells of the placenta. TSCs expressed fibronectin-binding, vitronectin-binding, and laminin-binding integrins, whereas the integrin ligands present in the TSC niche were collagen and laminin. Therefore, the only niche integrin ligand available for TSCs in vivo was laminin. Laminin promoted TSC adhesion and proliferation in vitro in an integrin binding–dependent manner. Importantly, when the integrin-binding ability of laminin was genetically ablated in mice, the size of the TSC population was significantly reduced compared with that in control mice. The present findings underscore an ECM niche function of laminin to support tissue stem cell maintenance in vivo.

## Introduction

Tissue stem cells maintain their ability to replicate and differentiate within a specialized microenvironment called the niche (Spradling et al, 2001). Stem cells require various soluble factors such as growth factors, morphogens, cytokines, and chemokines provided by the stem cell niche to maintain their undifferentiated state and self-renewal ability. In addition to these soluble factors, tissue stem cells require signals from the immobilized niche environment, that is, ECM to maintain their stemness. There are hundreds of ECM molecules encoded in the mammalian genome. These ECM molecules not only have diverse biological activities but also constitute supramolecular complexes that comprise the interstitial matrix and basement membrane. However, the diversity and complexity of ECMs in vivo make it difficult to decipher which ECM molecules contribute to stem cell maintenance as niche factors.

The placenta is the first organ that fixes embryos in the uterus and mediates physiological exchanges with the mother (Watson &

Cross, 2005). The tissue stem cells for the fetal placenta are trophoblast stem cells (TSCs) (Roberts & Fisher, 2011). Similar to other tissue stem cells, TSCs exist in their own niche. Specifically, TSCs first reside above the inner cell mass of the blastocyst and subsequently reside in the extraembryonic ectoderm (ExE) after implantation (Tanaka et al, 1998; Uy et al, 2002). TSCs represent a good model for investigation of niche functions in vivo because of the simple tissue constitution: the possible niche elements that surround TSCs in vivo comprise only the epiblast, endoderm, and basement membrane (Fig S1).

In the TSC niche, the epiblast provides the soluble factors FGF4 and nodal (Tanaka et al, 1998; Guzman-Ayala et al, 2004). FGF4 triggers phosphorylation of FGFR2 and formation of the GRB2/FRS2$\alpha$/SHP2 complex (Gotoh et al, 2005; Yang et al, 2006). In response to FGF4, FRS2$\alpha$ activates the ERK pathway to enhance the expression of CDX2. CDX2 is a transcription factor required for TSC establishment during ex vivo culture of embryos (Gotoh et al, 2005; Strumpf, 2005; Murohashi et al, 2010), but is dispensable for transdifferentiation of TSCs from fibroblasts (Kubaczka et al, 2015). Nodal or its related factors activin and TGF$\beta$ are required for maintenance of mouse TSCs in an undifferentiated proliferating state (Erlebacher et al, 2004; Guzman-Ayala et al, 2004). Inhibition of this signaling pathway leads to rapid down-regulation of CDX2 and FGFR2 expression (Erlebacher et al, 2004). Thus, although the niche functions of soluble factors are apparent, the kinds of ECM niche factors that regulate TSCs in vivo remain to be clarified.

In this study, we focused on the function of integrins because many ECM molecules are sensed by cell surface integrins. Integrins regulate various adhesion-dependent cellular behaviors, including cell migration, morphogenesis, proliferation, survival, and differentiation through binding to their ligands in ECMs (Legate et al, 2009). We examined the interactions between TSCs and their ECM niche via integrins and found that the only integrin ligand available for TSCs in vivo was laminin, the main component of the basement membrane. Laminin promoted TSC expansion in vitro, whereas nullification of its integrin-binding ability in vivo led to a significant decrease in the TSC population. These findings demonstrate the potency of laminin as the ECM niche for TSCs in vivo.

[1]Laboratory of Extracellular Matrix Biochemistry, Institute for Protein Research, Osaka University, Osaka, Japan  [2]Graduate School of Agricultural and Life Sciences, The University of Tokyo, Tokyo, Japan

Correspondence: sekiguch@protein.osaka-u.ac.jp
Daiji Kiyozumi's present address is Immunology Frontier Research Center, Osaka University, Osaka, Japan

# Results and Discussion

### Integrin expression profiles in TSCs

There are many integrin subtypes with distinct ligand specificities. To determine the integrin subtypes expressed in TSCs, integrin transcripts were quantified by real-time RT-PCR (Fig 1A). Because integrins are $\alpha/\beta$ heterodimeric receptors (Fig 1B) (Hynes, 2002; Barczyk et al, 2010), both types of subunits were investigated. A comprehensive survey of the transcript expressions for the integrin $\alpha$1–11, $\alpha$V, $\alpha$IIb, $\beta$1, and $\beta$3–8 subunits revealed that the major integrin subunits expressed in TSCs were $\alpha$3, $\alpha$5, $\alpha$6, $\alpha$7x1, $\alpha$7x2, $\alpha$V, $\beta$1, $\beta$3, $\beta$4, and $\beta$5 (Fig 1A). Given the $\alpha/\beta$ combinations known to date (Fig 1B), the integrin $\alpha\beta$ dimers expressed in TSCs were assumed to include laminin receptors ($\alpha$3$\beta$1, $\alpha$6$\beta$1, $\alpha$6$\beta$4, $\alpha$7x1$\beta$1, and $\alpha$7x2$\beta$1), fibronectin receptor ($\alpha$5$\beta$1), and vitronectin receptors ($\alpha$V$\beta$3 and $\alpha$V$\beta$5) (Fig 1C). Interestingly, TSCs expressed all of the laminin-binding integrins, but none of the collagen-binding integrins ($\alpha$1$\beta$1, $\alpha$2$\beta$1, $\alpha$10$\beta$1, and $\alpha$11$\beta$1) (Fig 1A and C).

To confirm that the obtained integrin expression profile matched the integrin function in TSCs, the cell–adhesive activities of TSCs toward ECM proteins defined as ligands for the individual integrin subtypes were examined. TSCs adhered to fibronectin, vitronectin, and laminin-111, but not to collagen types I and IV (Fig 1D), precisely reflecting the TSC integrin repertoire (Fig 1A and C).

### Integrin ligands in the TSC niche in vivo

We examined the ECM niche profile around TSCs by in situ integrin overlay assays, in which soluble recombinant integrins were able to bind and visualize their ligands in situ (Fig 2A) (Kiyozumi et al, 2012, 2014, 2018). We applied the assays to E5.5 embryonic sections because TSCs were shown to exist in the ExE at this developmental stage (Tanaka et al, 1998; Uy et al, 2002). A panel of recombinant integrins was applied to frozen E5.5 embryonic sections to examine their binding abilities toward the basement membrane of the ExE (indicated by cyan dotted lines in Fig 2B). The results revealed that integrin $\alpha$1$\beta$1 and $\alpha$7x2$\beta$1 bound extensively and integrin $\alpha$3$\beta$1 and $\alpha$10$\beta$1 bound less extensively to the ExE basement membrane, whereas no binding was observed for the other integrin subtypes (Fig 2B). Collectively, collagen-binding integrin $\alpha$1$\beta$1 and $\alpha$10$\beta$1 and laminin-binding integrin $\alpha$3$\beta$1 and $\alpha$7x2$\beta$1 bound to the ExE basement membrane (Fig 2B). These results indicate that collagen and laminin are available for TSCs as ECM niche components.

### Laminin-111 and laminin-511 are integrin ligands available for TSCs in vivo

By combining the integrin expression profile in TSCs (Fig 1C) with the survey results from the in situ integrin overlay assays (Fig 2B), the only integrin ligand available for TSCs at E5.5 was laminin (Fig 3A). According to a previous report (Miner et al, 2004), the laminin isoforms expressed in the E5.5 embryo are laminin-111 and laminin-511. We also confirmed that laminin $\alpha$1 and $\alpha$5 were expressed in the ExE basement membrane (Fig 3B and C). Because integrin $\alpha$7x2$\beta$1 binds to laminin-111 with high affinity and laminin-511 with moderate affinity (Nishiuchi et al, 2006; Kiyozumi et al, 2018), the strong in situ overlay signal with integrin $\alpha$7x2$\beta$1 on the ExE basement membrane (Fig 2B) indicates high integrin ligand activity of laminin-111 in situ. Meanwhile, the weak or no binding of $\alpha$3$\beta$1, $\alpha$6$\beta$4, and

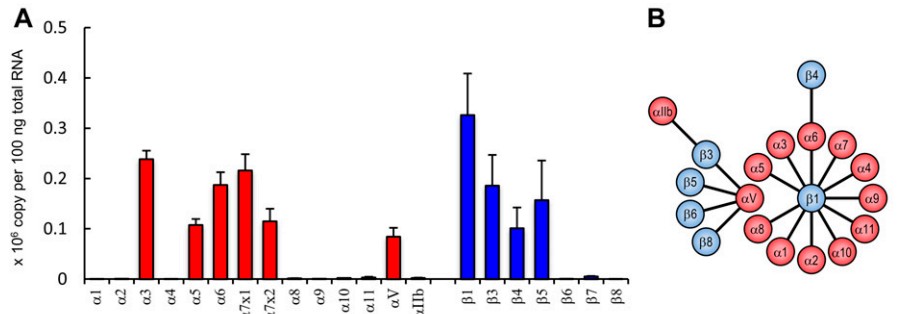

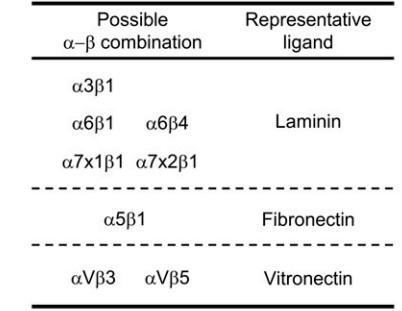

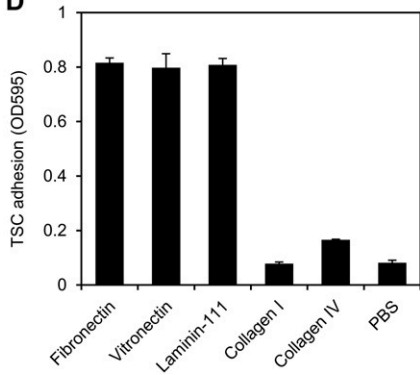

**Figure 1. Expression profile of integrin subunits in TSCs.**
**(A)** Transcript expression of integrin subunits in TSCs. Data represent means ± SD ($n$ = 3). **(B)** Diagrammatic representation of a known integrin $\alpha\beta$ heterodimer. The diagram is based on those in Hynes (2002) and Barczyk et al (2010). **(C)** Heterodimeric $\alpha\beta$ integrin subtypes expressed in TSCs. **(D)** Adhesion of TSCs to defined substrates. Data represent means ± SD ($n$ = 3).

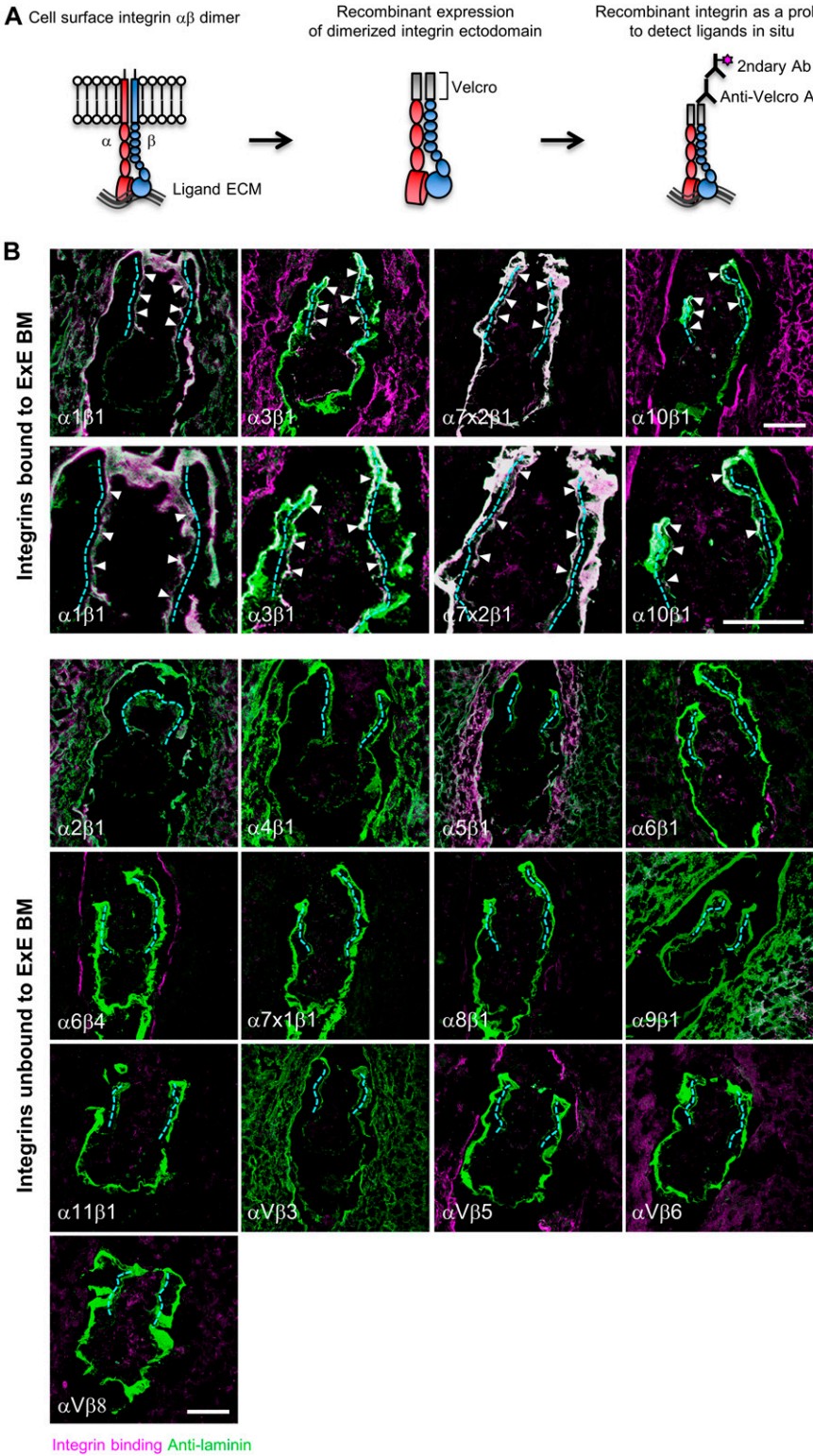

**Figure 2. Characterization of the ECM niche for TSCs in vivo.**
**(A)** Schematic views of in situ integrin ligand detection. Left, integrin $\alpha\beta$ dimer on the cell surface; middle, $\alpha\beta$ dimerized recombinant integrin ectodomain; right, in situ integrin ligand detection. **(B)** Comprehensive analyses of the ECM niche using E5.5 embryonic sections. Magenta, in situ binding of recombinant integrins; green, immunoreactivities for anti-laminin $\alpha1$ (for integrins $\alpha3\beta1$, $\alpha6\beta1$, $\alpha6\beta4$, $\alpha7x1\beta1$, $\alpha7x2\beta1$, $\alpha8\beta1$, $\alpha10\beta1$, $\alpha11\beta1$, $\alpha V\beta5$, $\alpha V\beta6$, and $\alpha V\beta8$) and anti-laminin $\gamma1$ (for integrins $\alpha1\beta1$, $\alpha2\beta1$, $\alpha4\beta1$, $\alpha5\beta1$, $\alpha9\beta1$, and $\alpha V\beta3$) mAbs; white, area double-positive for magenta and green signals; dotted lines, ExE basement membrane. The arrowheads indicate recombinant integrins bound to the ExE basement membrane. Bars, 50 $\mu$m.

$\alpha7x1\beta1$ integrins, all of which bind to laminin-511 (Nishiuchi et al, 2006; Kiyozumi et al, 2018), to the ExE basement membrane (Fig 2B) implies low availability of laminin-511 as the ligand for these integrin subtypes in situ, possibly because of its low expression level or a hitherto unknown modification that impairs its reactivity toward these integrin subtypes. The lack of binding of integrin $\alpha6\beta1$ (Fig 2B), which binds to both laminin-111 and laminin-511 (Nishiuchi et al, 2006; Kiyozumi et al, 2018), may be due to low sensitivity of its

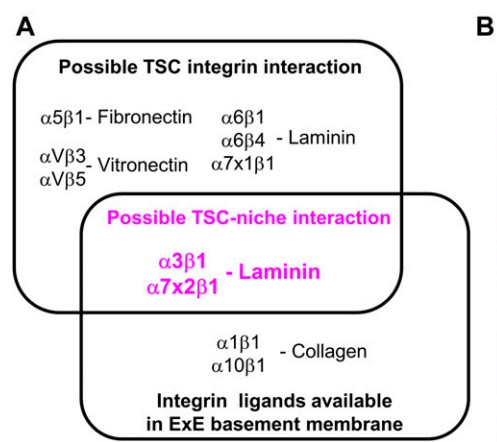

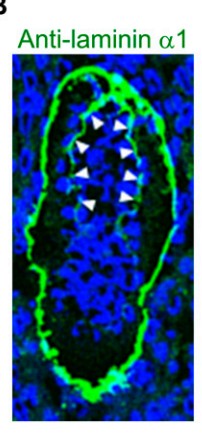

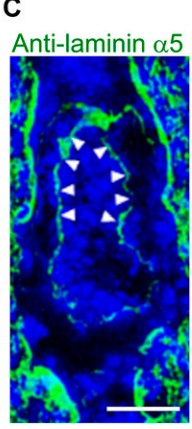

**Figure 3. The only integrin ligand available for TSCs is laminin.**
**(A)** Venn diagram representing possible TSC–integrin interactions, integrin ligands available in the ExE basement membrane, and possible TSC–niche interactions (magenta). **(B, C)** Localizations of the laminin α1 and α5 chains in E5.5 embryos. Wild-type E5.5 embryos were immunostained with anti-laminin α1 (B) and anti-laminin α5 (C) antibodies. Arrowheads, ExE basement membrane. Bar, 50 μm.

ligand detection under the conditions used. Human integrin α6β1 was shown to exhibit very weak affinity toward laminin-111 (Kiyozumi et al, 2018). When recombinant mouse integrin α6β1 was used in the in situ overlay assays, it gave positive signals on the ExE basement membrane (Fig S2A), consistent with the previous results. The signals for bound human integrin α3β1 or α7x2β1 as well as mouse integrin α6β1 were abolished in the presence of EDTA, confirming the divalent ion-dependent ligand binding of these integrins (Fig S2A and B). Taken together, we tentatively concluded that the integrin ligands practically available for TSCs in vivo are laminin-111 and laminin-511.

### Recombinant laminin supports in vitro TSC proliferation

We focused on laminin-111 as a major ECM niche for TSCs and investigated its function in supporting TSC adhesion and proliferation. As expected, TSCs adhered to Engelbreth-Holm-Swarm (EHS) mouse sarcoma-derived laminin-111 in a coating concentration-dependent manner (Fig 4A), endorsing the laminin receptor expression on TSCs (Fig 1C). Recombinant laminin-111 also potently promoted TSC adhesion (Fig 4A). The mutant recombinant laminin-111 EQ, which is inactive in integrin binding because of a Glu$^{1605}$ to Gln (EQ) point mutation in the laminin γ1 subunit (Ido et al, 2007; Kiyozumi et al, 2018), was very poor in promoting TSC adhesion (Fig 4A). These results indicate that laminin promotes TSC adhesion by interacting with integrin receptors. TSCs were also shown to adhere to laminin-511 and laminin-521 (Klaffky et al, 2001).

We further investigated whether laminin–integrin interactions promote TSC proliferation. TSCs can be cultured in medium supplemented with FGF4, heparin, and MEF-conditioned medium (MEF-CM) as a source of nodal/activin (Tanaka et al, 1998; Erlebacher et al, 2004). When TSCs were seeded onto laminin-111–coated dishes, the cells proliferated vigorously with a proliferation rate that was comparable with or faster than that under conventional TSC culture conditions (Fig 4B). TSC proliferation on laminin-111 was abolished when TSCs were seeded onto the integrin binding–inert laminin-111 EQ mutant (Fig 4B). These results indicate that the signals transduced by laminin–integrin interactions promote TSC proliferation. The ablated cell proliferation on laminin-111 EQ in the presence of FGF4 and MEF-CM further indicates that the integrin-mediated cell adhesion signals cannot be substituted by FGF or nodal/activin

signals and are a prerequisite for propagation of TSCs. Cell proliferation under conventional culture conditions on non-coated dishes likely occurred through adsorption of integrin ligands, such as fibronectin or vitronectin, derived from serum or MEF-CM onto culture dishes. TSCs can adhere to these substrates because they express fibronectin-binding and vitronectin-binding integrins (Fig 1). Indeed, the adhesion of TSCs onto 70CM-coated dishes was blocked by anti-β1, anti-β3, and anti-αV integrin antibodies (Fig S3).

Because supplementation of FGF4 with heparin and MEF-CM as the source of nodal/activin is indispensable for TSC proliferation (Tanaka et al, 1998; Erlebacher et al, 2004), we examined whether laminin-111 can substitute for the functions of FGF4 and MEF-CM by removing these factors from the culture medium. TSCs proliferated in the presence of FGF4 and MEF-CM on non-coated plates (Fig 4C). Removal of FGF4, heparin, and MEF-CM significantly suppressed TSC proliferation (Fig 4C), as reported previously (Tanaka et al, 1998; Erlebacher et al, 2004). Similarly, TSCs seeded onto laminin-111 in the absence of FGF4 and MEF-CM did not proliferate (Fig 4C), indicating that laminin-111 cannot substitute for FGF4 and MEF-CM. Collectively, these findings indicate that laminin–integrin interactions act together with soluble factors FGF4 and nodal/activin for TSC propagation.

It remains to be addressed whether TSCs cultured on laminin retain their differentiation potential. However, TSCs can be established from blastocysts under defined conditions by culture on Matrigel (Kubaczka et al, 2014). Because laminin is the major component of Matrigel, TSCs cultured on laminin may well retain their differentiation potential.

### Laminin–integrin interactions regulate TSC expansion in vivo

The above in vitro results raised the possibility that laminin supports TSC proliferation as an ECM niche factor in vivo. We recently developed the transgenic mouse line Lamc1$^{EQ}$, in which an EQ point mutation was introduced into the laminin-γ1 subunit. This laminin-γ1 EQ mutation abolishes the ability of laminins to bind to their cognitive integrins in vivo (Kiyozumi et al, 2018). For E5.5 embryos, the integrin-binding abilities of both laminin-111 and laminin-511 are nullified by the mutation. Thus, the mutant mouse line provides an opportunity to investigate the contribution of laminin–integrin interactions to TSC maintenance in vivo. In pre-implantation blastocysts,

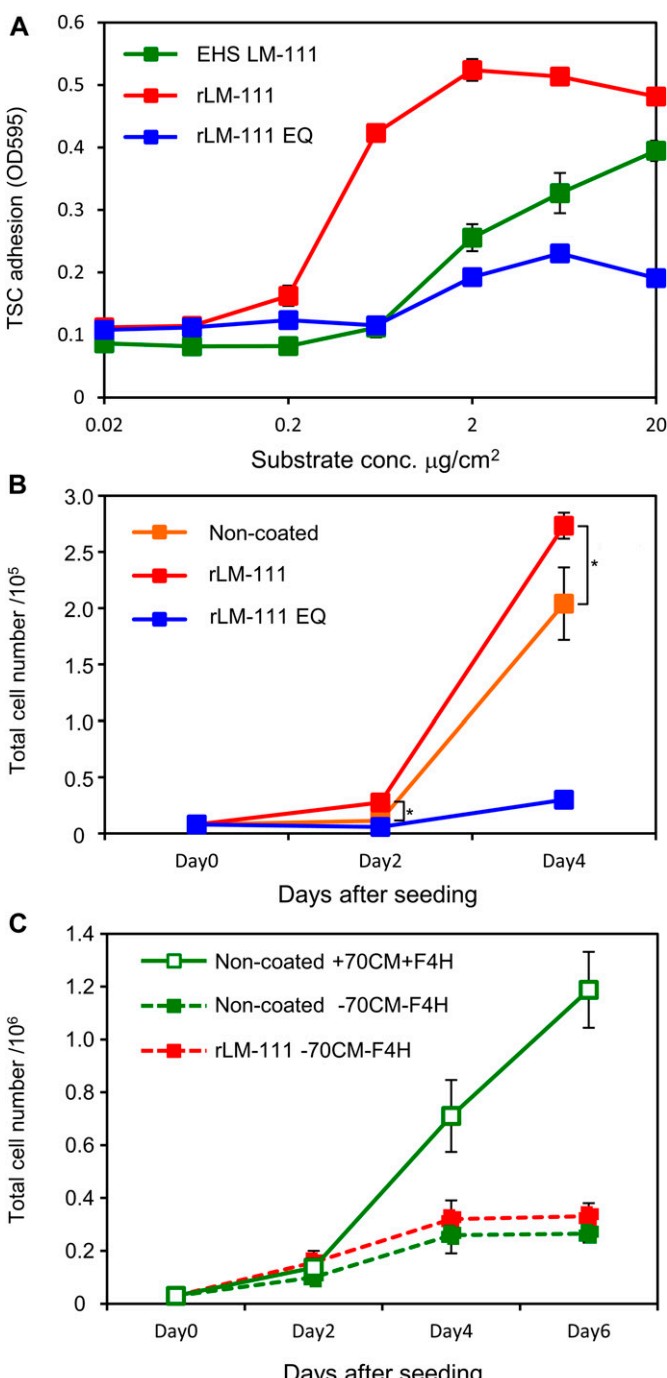

**Figure 4. TSCs proliferate on laminin-111.**
**(A)** TSC adhesion on various integrin ligands. Cell culture dishes were coated with increasing concentrations of EHS laminin-111 (EHS LM-111), recombinant laminin-111 (rLM-111), and recombinant laminin-111 EQ mutant (rLM-111 EQ). **(B)** TSC proliferation on non-coated, rLM-111–coated, and rLM-111 EQ–coated dishes. *$P < 0.05$, significant difference between non-coated and laminin-111–coated dishes by $t$ test. **(C)** TSC proliferation on non-coated or rLM-111–coated dishes. 70CM+F4H was added (Non-coated +70CM+F4H) or depleted (Non-coated –70CM–F4H and rLM-111 –70CM–F4H). Data represent means ± SD ($n = 3$).

both CDX2-positive trophoblast cells and OCT4-positive inner cell mass were observed (Fig 5A), indicating that early cell fate decisions were not affected in the laminin mutant mice. Because

embryogenesis is severely impaired in *Lamc1*$^{EQ/EQ}$ homozygotes until E7.5 (Kiyozumi et al, 2018), the TSC population after implantation was investigated in *Lamc1*$^{EQ/EQ}$ homozygotes at E5.5. In the E5.5 egg cylinder of *Lamc1*$^{EQ/EQ}$ mice, recombinant integrin binding to laminin in situ was abolished (Fig 5B), confirming the ablation of laminin–integrin interactions in these mutant mice. At E5.5, CDX2-positive cells were observed but became detached from the laminin-positive basement membrane (Fig 5C), endorsing our conclusion that the only integrin ligand available for TSCs at E5.5 is laminin. Importantly, the population size of CDX2-positive cells in *Lamc1*$^{EQ/EQ}$ homozygotes was decreased compared with that in control littermates (Figs 5C and S4). Quantitative image analyses confirmed the significant decrease in CDX2-positive cells in *Lamc1*$^{EQ/EQ}$ mice compared with control littermates (Fig 5D), whereas OCT4-positive epiblast cells were not critically affected (Fig 5C and D). These results indicate that γ1 chain–containing laminin supports TSCs as an ECM niche factor and that, in the absence of laminin–integrin interactions, TSCs cannot expand their population in vivo.

TSCs did not express collagen-binding integrins (Fig 1A). However, TSCs can be established from human blastocysts cultured on collagen type IV in vitro (Okae et al, 2018). It was also reported that collagen type IV can induce trophoectoderm differentiation of mouse embryonic stem cells in vitro (Schenke-Layland et al, 2007). Given that non-integrin collagen receptors exist (Ricard-Blum, 2011), collagens may contribute to the ECM niche for TSCs in an integrin-independent manner.

In several tissue stem cells, such as spermatogonial stem cells, neural stem cells, and mammary stem cells, laminin–integrin interactions have been proposed to function in the regulation of stem cell behaviors, including homing and maintenance in the niche (Kanatsu-Shinohara et al, 2008; Nascimento et al, 2018; Romagnoli et al, 2019; Sato et al, 2019). The present results allow us to clearly conclude that laminin is the in vivo ECM niche for TSCs and functions by interacting with cognitive integrins, as modeled in Fig 5E. Further investigations on the molecular basis of laminin function as a niche for various tissue stem cells would facilitate better understanding of how stem cell behaviors are regulated by the ECM niche as well as the development of new approaches for stem cell–based regenerative therapy using ECMs.

## Materials and Methods

### Mice

The *Lamc1*$^{EQ}$ knock-in mouse line was generated in a previous study (Kiyozumi et al, 2018). Genotyping of *Lamc1*$^{EQ}$ mice was performed by genomic PCR or in situ integrin binding as described (Kiyozumi et al, 2018).

The mice were maintained in a specific pathogen-free environment under stable conditions of temperature (25°C) and light (lights on at 08:00 and off at 20:00). All mouse experiments were performed in compliance with our institutional guidelines and were approved by the Animal Care Committee of Osaka University.

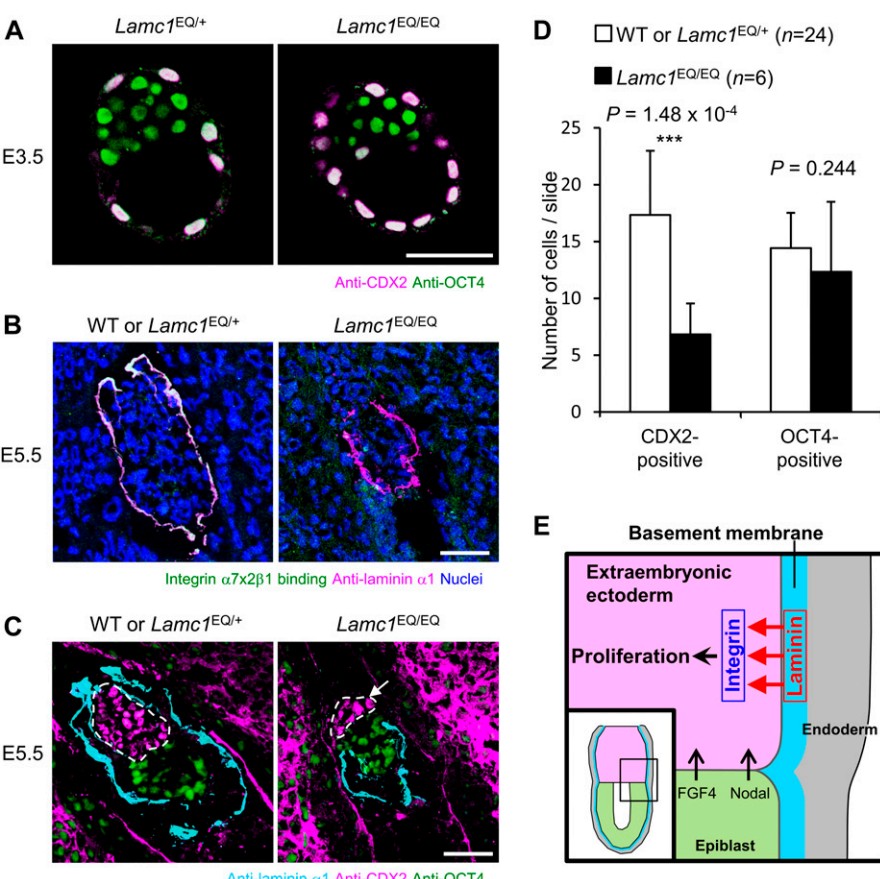

**Figure 5. Laminin–integrin interactions contribute to TSC expansion in vivo.**
**(A)** Expression of undifferentiated TSC marker CDX2 in trophectoderm cells in *Lamc1*[EQ/+] and *Lamc1*[EQ/EQ] blastocysts. Bar, 50 *µm*. **(B)** Loss of integrin α7x2β1 binding to the basement membrane in E5.5 *Lamc1*[EQ/EQ] embryos. Magenta, anti-laminin α1 antibody binding; green, recombinant integrin α7x2β1 binding; white, area double-positive for magenta and green signals. Bar, 50 *µm*. **(C)** Morphologies of E5.5 control and *Lamc1*[EQ/EQ] embryos visualized by immunofluorescence. Cyan, laminin α1; magenta, CDX2; green, OCT4. ExE cells are enclosed by dotted lines. The arrow indicates CDX2-positive cells detached from the laminin-positive basement membrane. Bar, 50 *µm*. **(D)** Quantification of CDX2-positive and OCT4-positive cells in control WT and *Lamc1*[EQ/+] E5.5 egg cylinders. Data represent means ± SD (*n* = 24 and 6 for WT or *Lamc1*[EQ/+] and *Lamc1*[EQ/EQ], respectively). ***P < 0.001, significant difference by Welch's *t* test. **(E)** Diagram illustrating the dependence of TSCs on laminin–integrin interactions in the mouse conceptus. In addition to FGF4 and nodal from the epiblast, laminin also acts on TSCs as an ECM niche through binding to integrin receptors. The inset shows the region illustrated in the main figure. The diagram is based on that in Tanaka et al (1998).

## Antibodies and reagents

Rat mAb against mouse laminin α1 (5B7-H1), rat mAb against mouse laminin α5 (M5N8-C8), and rabbit polyclonal antibody (pAb) against Velcro (ACID/BASE coiled-coil) peptides were produced in previous studies (Manabe et al, 2008; Sato-Nishiuchi et al, 2012). The following antibodies and reagents were obtained commercially: rat anti-laminin-γ1 mAb (Millipore); rabbit anti-laminin pAb, BSA, and heparin (Sigma-Aldrich); mouse anti-CDX2 mAb (Biocare); rabbit anti-OCT4 pAb (Santa Cruz Biotechnology); rat anti-integrin αV mAb (RMV-7), hamster anti-integrin β1 mAb (Ha2/5), and hamster anti-integrin β3 mAb (2C9.G2) (BD Biosciences); Alexa 488–conjugated goat antirabbit IgG, Alexa 546–conjugated goat antirat IgG, Alexa 405–conjugated goat antirabbit IgG, Alexa 488–conjugated goat antirat IgG, and Alexa 546–conjugated goat antimouse IgG (Invitrogen); bovine types I and IV collagens (Nippi Inc.); recombinant human FGF4 (Peprotech); and PermaFluor (Thermo Shandon). Mouse EHS laminin was prepared from a mouse EHS tumor as described (Murayama et al, 1996). Fibronectin was purified from human plasma by gelatin affinity chromatography as described (Sekiguchi & Hakomori, 1983).

## Expression vectors

Expression vectors for the extracellular domains of the human integrin α1, α2, α3, α6, α7x1, α7x2, α8, and α9 subunits were constructed as described (Nishiuchi et al, 2006; Sato et al, 2009; Sato-Nishiuchi et al, 2012; Jeong et al, 2013; Ozawa et al, 2016; Kiyozumi et al, 2018). Expression vectors for the extracellular domains of the human integrin α5, αV, β1, β3, and β4 subunits were kindly provided by Dr Junichi Takagi (Institute for Protein Research, Osaka University) (Takagi et al, 2001, 2002a, 2002b). Expression vectors for the extracellular domains of the human integrin α4, α10, α11, β6, and β8 subunits were constructed as described (Kiyozumi et al, 2014). Expression vectors for the laminin β1, γ1, and γ1 EQ fragments were constructed as described (Ido et al, 2007). An expression vector for the His[6]-tagged laminin α1 fragment was constructed by cloning a cDNA corresponding to amino acids Phe[1878]–Ser[3675] as described (Ido et al, 2007).

## Expression and purification of recombinant proteins

Recombinant human laminin-111 fragment and its EQ mutant were produced using a FreeStyle 293 Expression System (Thermo Fisher Scientific) as described (Ido et al, 2004). The conditioned media were passed over Anti-FLAG M2 Affinity Gel (Sigma-Aldrich). After washing with 20 mM TBS without divalent cations, bound proteins were eluted with 100 *µ*g/ml FLAG peptide (Sigma-Aldrich) and dialyzed against TBS. Other recombinant proteins (human integrins α1β1, α2β1, α3β1, α4β1, α5β1, α6β1, α7x1β1, α7x2β1, α8β1, α9β1, α10β1, α11β1, αVβ3, αVβ5, αVβ6, and αVβ8; mouse integrin α6β1 and vitronectin) were produced using the FreeStyle 293 Expression System and affinity-purified as described (Kiyozumi et al, 2014, 2018;

Ozawa et al, 2016). Protein concentrations were determined with a BCA Protein Assay Kit (Thermo Fisher Scientific) using BSA as the standard.

### In situ integrin binding

In situ integrin binding was performed as described previously (Kiyozumi et al, 2012, 2014). Frozen sections of E5.5 mouse embryos were blocked with blocking buffer (3% BSA, 25 mM Tris–HCl, pH 7.4, 100 mM NaCl, and 1 mM $MnCl_2$) for 30 min at room temperature and incubated with 3 $\mu$g/ml recombinant integrin and rat anti-laminin $\alpha$1 mAb (for integrins $\alpha$3$\beta$1, $\alpha$6$\beta$1, $\alpha$6$\beta$4, $\alpha$7x1$\beta$1, $\alpha$7x2$\beta$1, $\alpha$8$\beta$1, $\alpha$10$\beta$1, $\alpha$11$\beta$1, $\alpha$V$\beta$5, $\alpha$V$\beta$6, and $\alpha$V$\beta$8) or rat anti-laminin-$\gamma$1 mAb (for integrins $\alpha$1$\beta$1, $\alpha$2$\beta$1, $\alpha$4$\beta$1, $\alpha$5$\beta$1, $\alpha$9$\beta$1, and $\alpha$V$\beta$3) in the presence or absence of 10 mM EDTA in blocking buffer at 4°C overnight. The sections were washed three times with wash buffer (25 mM Tris–HCl, pH 7.4, 100 mM NaCl, and 1 mM $MnCl_2$) for 10 min at room temperature, and incubated with 0.5 $\mu$g/ml rabbit anti-Velcro pAb in blocking buffer at room temperature for 2 h. After washing with wash buffer, the sections were incubated with Alexa 488–conjugated goat antirabbit IgG and Alexa 546–conjugated goat antirat IgG. The nuclei were stained with Hoechst 33342 if necessary. After washing with wash buffer, the sections were mounted in PermaFluor and visualized with an LSM510 laser confocal microscope (Carl Zeiss).

### Real-time RT-PCR

Total RNA was extracted from TSCs with an RNeasy Kit (QIAGEN) and cDNA was synthesized using SuperScript III reverse transcriptase (Invitrogen) according to the manufacturer's instructions. The primers used for RT-PCR are shown in Table S1. RT products corresponding to 100 ng of total RNA were used for each PCR. Real-time RT-PCR was performed with a Smart Cycler (Cepheid).

### TSCs

TSCs established from a mouse blastocyst were maintained under a feeder-free condition based on an established culture protocol (Hayakawa et al, 2015). TS medium was prepared by supplementing RPMI 1640 medium with 20% FBS, 100 $\mu$M 2-mercaptoethanol, 2 mM L-glutamine, 1 mM sodium pyruvate, 50 U/ml penicillin, and 50 mg/ml streptomycin. MEF-CM was prepared by culturing mitomycin C–treated MEFs in TS medium. Seventy percent CM (70CM) was prepared by mixing by TS medium and MEF-CM at a ratio of 3:7. TSCs were maintained in 70CM supplemented with 25 $\mu$g/ml FGF4 and 1 $\mu$g/ml heparin (70CM+F4H).

### Cell adhesion assay

Ninety six–well cell culture dishes were coated with 50 $\mu$l of ECM molecules at 15 $\mu$g/ml or other specified concentrations in PBS or 70CM. TSCs were suspended in RPMI 1640 medium containing 10 mg/ml BSA, 1 mM glutamine, 1 mM sodium pyruvate, and 10 mM Hepes (pH 7.4), and the cell density was adjusted to 6 × $10^5$ cells/ml. Fifty microliters of TSC suspension was seeded in the plates and incubated for 1 h at 37°C. For integrin inhibition, 10 $\mu$g/ml of various anti-integrin antibodies was included in the TSC suspension. After removal of unattached cells by washing with cell suspension medium, the attached cells were fixed with 3.7% formalin in PBS and stained with 0.1% toluidine blue in PBS. After lysis in 1% SDS, the attached cells were quantified by the absorption at 595 nm. An OD value of 0.8 corresponded to ~50% occupancy of the culture dish surface by the seeded cells.

### Cell proliferation assay

Twenty four–well cell culture dishes were coated with various ECM molecules at a density of 2 $\mu$g/$cm^2$ and seeded with TSCs at a density of 0.8 × $10^4$ cells/well. The cells were cultured in 70CM+F4H, and the cell numbers were counted every 2 d. For FGF4/MEF-CM depletion assays, TSCs were seeded at a density of 3 × $10^4$ cells/well and cultured under a feeder-free condition in the presence or absence of 70CM+F4H.

### Immunofluorescence

Frozen sections of mouse embryos were fixed with 4% paraformaldehyde in PBS at 4°C for 10 min, blocked with 3% BSA in 0.1% Tween-20/PBS (TPBS) at 4°C for 30 min, and incubated at 4°C overnight with antibodies diluted in 3% BSA/TPBS. The sections were washed three times with TPBS for 10 min, and incubated at 4°C for 2 h with the following secondary antibodies: Alexa 488–conjugated goat antirabbit IgG and Alexa 546–conjugated goat antirat IgG; or Alexa 405–conjugated goat antirabbit IgG, Alexa 488–conjugated goat antirat IgG, and Alexa 546–conjugated goat antimouse IgG. The nuclei were stained with Hoechst 33342 if necessary. After three washes with TPBS for 10 min, the sections were mounted in PermaFluor. Whole-mount immunofluorescence of blastocysts was performed as described previously (Kiyozumi et al, 2018). The fixed blastocysts were washed with PBS, permeabilized with 0.1% Triton X-100 in PBS at 4°C for 10 min and incubated with mouse anti-CDX2 mAb and rabbit anti-OCT4 pAb diluted in TPBS at 4°C overnight. The embryos were washed three times with TPBS for 10 min and incubated with Alexa 546–conjugated goat antirabbit IgG and Alexa 488–conjugated goat antirat IgG. After three washes with TPBS for 10 min, the blastocysts were placed in a small drop of PBS covered with mineral oil on a coverslip. All specimens were visualized using the LSM510 laser confocal microscope.

The LSM510 laser confocal microscope was equipped with LD-Achroplan (20×, NA 0.4) and Plan-Neofluar (40×, NA 0.75) objective lenses and operated at room temperature. The imaging medium was air. The LSM510 PASCAL software (Carl Zeiss) was used for image collection. Each set of stained samples was processed under identical gain and laser power settings. Each set of obtained images was processed under identical brightness and contrast settings, which were adjusted by the LSM image browser (Carl Zeiss) for clear visualization of immunostaining.

### Image analysis

The numbers of CDX2-positive or OCT4-positive cells were counted on immunofluorescence-stained images of E5.5 egg cylinders.

## Statistical analysis

The statistical significance of differences in data was determined by a two-tailed $t$ test or Welch's $t$ test using Microsoft Excel for Mac 2011. Values of $P < 0.001$ or $P < 0.05$ were considered to indicate statistical significance.

## Supplementary Information

## Acknowledgements

We thank Drs Yukimasa Taniguchi, Chisei Shimono, Yuya Sato, and Akio Ozawa for the expression and purification of recombinant integrins. We also thank Alison Sherwin, PhD, from Edanz Group (www.edanzediting.com/ac) for editing a draft of this manuscript. This work was supported by KAKENHI grants 17082005 and 22122006 (to K Sekiguchi).

### Author Contributions

D Kiyozumi: conceptualization, investigation, and methodology.
I Nakano: methodology.
R Sato-Nishiuchi: resources.
S Tanaka: methodology.
K Sekiguchi: conceptualization and resources.

### Conflict of Interest Statement

The authors declare that they have no conflict of interest.

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
