## [Reviewer comments · Life Science Alliance]

LAMININ IS THE ECM NICHE FOR TROPHOBLAST STEM CELLS

Daiji Kiyozumi, Itsuko Nakano, Ryoko Sato-Nishiuchi, Satoshi Tanaka, and Kiyotoshi Sekiguchi

DOI: 10.26508/lsa.201900515

Corresponding author(s): Kiyotoshi Sekiguchi, Osaka University

Review timeline:

Submission Date:	2019-08-05
Editorial Decision:	2019-09-12
Revision Received:	2019-12-11
Editorial Decision:	2019-12-31
Revision Received:	2020-01-06
Accepted:	2020-01-07

Scientific Editor: Andrea Leibfried

Transaction Report:

1st Editorial Decision

12 September 2019

September 12, 2019

Re: Life Science Alliance manuscript #LSA-2019-00515-T

Prof. Kiyotoshi Sekiguchi
Osaka University
Institute for Protein Research
3-2 Yamadaoka
3-2 Yamadaoka
Suita, Osaka 565-0871
Japan

Dear Dr. Sekiguchi,

Thank you for submitting your manuscript entitled "LAMININ IS THE ECM NICHE FOR TROPHOBLAST STEM CELLS" to Life Science Alliance. The manuscript was assessed by expert reviewers, whose comments are appended to this letter.

As you will see, the reviewer raise some concerns and point to lacking information, the need for clarifications and the need to add some controls. Given this input, we would like to invite you to submit a revised version of your manuscript to us, addressing the concerns of the reviewers.

Thank you for this interesting contribution to Life Science Alliance. We are looking forward to receiving your revised manuscript.

Sincerely,

Andrea Leibfried, PhD
Executive Editor
Life Science Alliance
Meyerhofstr. 1

69117 Heidelberg, Germany
t +49 6221 8891 502
e a.leibfried@life-science-alliance.org
www.life-science-alliance.org

B. MANUSCRIPT ORGANIZATION AND FORMATTING:

Reviewer #1 (Comments to the Authors (Required)):

The manuscript deals with the nature of the extracellular niche of trophoblast stem cells (TSCs) during blastocyte implantation. The authors have addressed this question from the perspective of the integrin receptors - a novel and interesting aspect. However, the nature of the specific integrins involved is not clearly shown by the data presented and the question of the extracellular matrix (ECM) substrate is also unclear. As laminins have already been shown to be critical at this stage of mouse development, the new data provided by the manuscript is limited. In addition, and some explanations of experimental procedures is required and potentially extra control experiments.

In Figure 1 what does an OD of 0.8 reflect in terms of total cells bound - is it 10%, 50% or 100% of the total cells added? This would aid in the understanding of the

addictiveness of the cells in general. This also is relevant for the adhesion data shown in Figure 4A.

Interesting overlay integrin assays are employed, where tagged recombinant/integrin dimers are added to tissue sections and then visualized with antibodies to the tags - this overcomes the limitations in the availability of antibodies to mouse integrin subunits. The authors refer to weak and strong binding of the recombinant integrins to tissue sections - this cannot be stated from these types of experiments, rather extensive or less extensive. In general, however, the data in Figure 2B is not clear - it may be necessary to show the integrin binding separately. What is the white staining? Why is the green anti-laminin staining so different in some sections eg where integrin $\alpha 7\beta 1$, $\alpha 6\beta 4$, $\alpha 7\beta 1$, $\alpha 8\beta 1$, $\alpha 11\beta 1$, $\alpha v\beta 5$ and $\alpha v\beta 6$ and $\alpha v\beta 8$ are also tested - either there is no staining or just Reichert's BM is stained and not the vessels of the surrounding tissue? What does 'Integrins bound' and 'Integrins unbound' refer to in this figure? As this data is critical for the rest of the manuscript, it is important to clearly show the bound recombinant integrins and to provide appropriate controls for these experiments (please see below).

By deduction of integrins expressed on the TSCs and the differential binding of the recombinant integrins to the tissue sections the authors propose $\alpha 3\beta 1$ and $\alpha 7\beta 1$ mediated interactions between TSCs and laminin 111 in the tissue. The absence of binding of integrin $\alpha 6\beta 1$, however, is not consistent with this hypothesis as $\alpha 6\beta 1$ is one of the highest affinity receptors for laminin 111. The authors state that the absence of binding of the $\alpha 6\beta 1$ construct may be due to the conditions used for the overlay experiments, but this argument applies to all other integrins as well and raises the question of how reliable and specific such overlay assays are. Some controls would be of advantage here e.g. the binding of the recombinant integrins under different conditions such as without Ca^{2+} and Mg^{2+} to ablate all binding, and in the presence of different cations to promote binding of specific integrins, or with Mn^{2+} to promote binding of all constructs.

Figure 4: Can the TSCs also bind to laminin 511 and to laminin 332 (as a control)? While the experiments performed with laminin 111 in which the Glu1605, which is required for integrin binding, is replaced with Gln show that integrins are involved, this does not support binding of specifically $\alpha 7\beta 1$ or $\alpha 3\beta 1$, as this residue is also required for $\alpha 6\beta 1$ binding to laminin 111 and 511.

Are the differences in proliferation of TSCs on laminin 111 compared to uncoated dishes statistically significant? The authors explain the high proliferation on uncoated plastic as binding of the cells to fibronectin or vitronectin for serum from the cell culture medium - can this binding be blocked for example with RGD, anti-integrin $\alpha 5$ or anti-integrin $\beta 1$ Ab? Inclusion of such controls on both the cells plated on uncoated and laminin 111 coated dishes would strengthen the data.

Labelling of immunofluorescence images in Figure 5 need to be clearer - here too it is not clear what the white staining is in the left panel of B (presumably laminin 1).

Reviewer #2 (Comments to the Authors (Required)):

In this report, the authors investigate a role for laminin in the trophoblast stem cell (TSC) niche. The authors utilize a variety of research approaches to examine laminin-integrin signaling in the early mouse embryo. The research provides an important scientific advance. Some concerns with the experimental design and interpretation of results are presented below..

1. Page 5, the comment that Cdx2 is indispensable for establishment of TSCs is possibly an overreach. Reprogramming experimental results published in Cell Stem Cell in 2015 demonstrated that Cdx2 was not needed in the transcription factor cocktail to generate TSCs.

2. The integrin binding assay requires some additional details, especially the method of detecting integrin binding. This was apparently accomplished with the anti-Velcro antibody. What is "Velcro"? What does the anti-Velcro antibody recognize? What is the source of the anti-velcro antibody?

3. The authors provide information about laminin promoting adhesion and proliferation of TSCs. Do the TSCs cultured on laminin retain their abilities to differentiate into trophoblast giant cells, and other differentiated trophoblast lineages? Some evidence about the capacity for trophoblast differentiation would be helpful.

4. In the literature there is some evidence for the importance of IV collagen as a contributor to the TSC niche. See the following reports:

Schenke-Layland K, Angelis E, Rhodes KE, Heydarkhan-Hagvall S, Mikkola HK, Maclellan WR. Collagen IV induces trophoectoderm differentiation of mouse embryonic stem cells. *Stem Cells*. 2007 Jun;25(6):1529-38. Epub 2007 Mar 15. PubMed PMID: 17363553.

Okae H, Toh H, Sato T, Hiura H, Takahashi S, Shirane K, Kabayama Y, Suyama M, Sasaki H, Arima T. Derivation of Human Trophoblast Stem Cells. *Cell Stem Cell*. 2018 Jan 4;22(1):50-63.e6. doi: 10.1016/j.stem.2017.11.004. Epub 2017 Dec 14. PubMed PMID: 29249463.

Some comments may be helpful.

5. The culture medium used in the in vitro analysis is important, especially the presence of FBS. Information about the generation of the MEF conditioned culture medium should be provided. Was FBS in the MEF culture medium?

6. Fig. 1: The integrin profile in cultured TSCs is useful. What is the integrin profile of a trophoblast stem cell developing in situ? What is the basis of the assumption that the culture conditions for TSCs fully replicate the in vivo environment?

7. Fig. 2: It is not clear how the integrin binding was detected.

8. Fig. 4: Do the ECM proteins used in the cell adhesion or proliferation assays contain contaminating cytokines or growth factors? What is the basis for determining purity of the ECM protein preparations? If contaminants are present, then could they influence TSC proliferation?

9. Fig. 4: In panel C, rLM-111 +70CMF4H should be included as a positive control to compare with the rLM-111 -70CMF4H.

10. Fig. 5, panel C: What is the specificity of the Cdx2 antibody? It looks there is extensive Cdx2 immunoreactivity throughout the adjacent uterine decidua.

Reviewer #3 (Comments to the Authors (Required)):

The authors have examined the role of extracellular matrix in trophoblast stem cell (TSC) biology. They have concluded that laminin-111 interaction with integrin alpha7X2beta1 plays a critical role in the adhesion, growth and maintenance of these

cells based on both in vitro and in vivo experiments.

Overall the study has been carefully carried out with the major findings supported by the data. By determining the integrin and laminin subunits expressed at the trophoblast basement membrane (BM) zone and the integrins that bind to the BM, the authors were able to deduce the major interaction that occur in situ. In addition, the authors show that laminin-111 supports TSC adhesion and proliferation in the presence of F4H and heparin. Particularly persuasive is the comparison made between wild-type and laminin-gamma1 E:Q mutant mouse embryos, the latter unable to bind to integrins, in which it was found that the population of TSC cells is reduced in the absence of the specific integrin binding to the laminin.

Specific questions (minor concerns):

1. Fig. 2. The immunofluorescence colors are magenta (integrin) and green (laminin). In the methods, Alexa 488 (green) is listed as the reagent used to detect applied integrin via the anti-Velcro antibody while Alexa 546 (yellow/orange) is used to detect the laminin. This is a bit confusing. Have the authors used pseudo color assignments in the figure as shown?
2. Fig. 2. It would help the reader to see larger images of the region of the extraembryonic BM.
3. page 8, upper paragraph. Integrin alpha7x2beta1. Is there a phenotype in the integrin alpha7 knockout mouse known to correspond to an alteration of trophoblast stem cells?

The original manuscript has been revised in accordance with your suggestions and those from the three reviewers. Our detailed point-by-point responses to the reviewers' comments are provided below. Ryoko Sato-Nishiuchi, who produced recombinant mouse integrin $\alpha 6\beta 1$, was included as a coauthor in the revised manuscript, because the mouse integrin was an essential resource in the in situ integrin overlay assays shown in the new **Supplementary Figure S2**.

Reviewer #1

The manuscript deals with the nature of the extracellular niche of trophoblast stem cells (TSCs) during blastocyte implantation. The authors have addressed this question from the perspective of the integrin receptors - a novel and interesting aspect. However, the nature of the specific integrins involved is not clearly shown by the data presented and the question of the extracellular matrix (ECM) substrate is also unclear. As laminins have already been shown to be critical at this stage of mouse development, the new data provided by the manuscript is limited. In addition, some explanations of experimental procedures is required and potentially extra control experiments.

In Figure 1 what does an OD of 0.8 reflect in terms of total cells bound - is it 10%, 50% or 100% of the total cells added? This would aid in the understanding of the addictiveness of the cells in general. This also is relevant for the adhesion data shown in Figure 4A.

- An OD of 0.8 corresponded to ~50% occupancy of the culture dish surface by the seeded cells. We have added this information to the **Materials and Methods** (page 17, lines 11–12, revised manuscript).

Interesting overlay integrin assays are employed, where tagged recombinant/integrin dimers are added to tissue sections and then visualized with antibodies to the tags - this overcomes the limitations in the availability of antibodies to mouse integrin subunits. The authors refer to weak and strong binding of the recombinant integrins to tissue sections - this cannot be stated from these types of experiments, rather extensive or less extensive. In general, however, the data in Figure 2B is not clear - it may be necessary to show the integrin binding separately. What is the white staining? Why is the green anti-laminin staining so different in some sections eg where integrin $\alpha 7x2\beta 1$, $\alpha 6\beta 4$, $\alpha 7x1\beta 1$, $\alpha 8\beta 1$, $\alpha 11\beta 1$, $\alpha v\beta 5$ and $\alpha v\beta 6$ and $\alpha v\beta 8$ are also tested - either there is no staining or just Reichert's BM is stained and not the vessels of the surrounding tissue? What does 'Integrins bound' and 'Integrins unbound' refer to in this figure? As this data is critical for the rest of the manuscript, it is important to clearly show the bound recombinant integrins and to provide appropriate controls for these experiments (please see below).

- In accordance with the reviewer's suggestion, we have amended the description "integrin $\alpha 1\beta 1$ and $\alpha 7x2\beta 1$ strongly, and $\alpha 3\beta 1$ and $\alpha 10\beta 1$ weakly, bound to the ExE basement membrane" to "integrin $\alpha 1\beta 1$ and $\alpha 7x2\beta 1$ bound extensively and integrin $\alpha 3\beta 1$ and $\alpha 10\beta 1$ bound less extensively" (page 7, lines 8–9, revised manuscript).
- The white color indicates the area double-positive for anti-laminin antibody binding (magenta) and recombinant integrin binding (green), because white is the consequence of superimposing magenta and green. We have included this information in the legend for **Figure 2B** in the revised manuscript.
- We used anti-laminin $\alpha 1$ mAb (for integrins $\alpha 3\beta 1$, $\alpha 6\beta 1$, $\alpha 6\beta 4$, $\alpha 7x1\beta 1$, $\alpha 7x2\beta 1$, $\alpha 8\beta 1$, $\alpha 10\beta 1$, $\alpha 11\beta 1$, $\alpha V\beta 5$, $\alpha V\beta 6$, and $\alpha V\beta 8$) or anti-laminin $\gamma 1$ mAb (for integrins

- $\alpha 1\beta 1$, $\alpha 2\beta 1$, $\alpha 4\beta 1$, $\alpha 5\beta 1$, $\alpha 9\beta 1$, and $\alpha V\beta 3$) to visualize the extraembryonic (ExE) basement membrane. We have added this information to the **Materials and Methods** (page 15, lines 15–19, revised manuscript). Anti-laminin $\alpha 1$ mAb visualizes the zygotic basement membrane only, while anti-LN $\gamma 1$ mAb visualizes both the zygotic and maternal basement membranes. Both antibodies bind to the ExE basement membrane but their signal strengths vary in the surrounding maternal tissues.
- “Integrins bound” refers to “integrin isoforms bound to the ExE basement membrane” as indicated by the arrowheads in **Figure 2B** ($\alpha 1\beta 1$, $\alpha 3\beta 1$, $\alpha 7x2\beta 1$, $\alpha 10\beta 1$). We have rephrased the captions for improved clarity.
 - The specificity of the *in situ* integrin overlay assays was repeatedly confirmed in our preceding reports (Kiyozumi et al., *Curr. Protoc. Cell Biol.* 2014;65:10.19.1-10.19.17; Kiyozumi et al., *J. Cell Biol.* 2012;197:677-689; Sato-Nishiuchi et al., *J. Biol. Chem.* 2012;287:25615-25630), based on nullification of signals for bound integrins when divalent cations were depleted by EDTA. To address this point, we have added data showing that the signals for bound integrins were nullified in the presence of EDTA (**Supplementary Figure S2** and page 8, lines 13–16, revised manuscript).

By deduction of integrins expressed on the TSCs and the differential binding of the recombinant integrins to the tissue sections the authors propose $\alpha 3\beta 1$ and $\alpha 7x2\beta 1$ mediated interactions between TSCs and laminin 111 in the tissue. The absence of binding of integrin $\alpha 6\beta 1$, however, is not consistent with this hypothesis as $\alpha 6\beta 1$ is one of the highest affinity receptors for laminin 111. The authors state that the absence of binding of the $\alpha 6\beta 1$ construct may be due to the conditions used for the overlay experiments, but this argument applies to all other integrins as well and raises the question of how reliable and specific such overlay assays are. Some controls would be of advantage here e.g. the binding of the recombinant integrins under different conditions such as without Ca^{2+} and Mg^{2+} to ablate all binding, and in the presence of different cations to promote binding of specific integrins, or with Mn^{2+} to promote binding of all constructs.

- As described above, we repeatedly confirmed the specificity of the *in situ* integrin overlay assays, by performing the assays in parallel with and without 10 mM EDTA (page 8, lines 13–16 and page 15, line 18, revised manuscript).
- Unlike the results for the cell adhesion assays and cell adhesion inhibition assays with the anti-integrin $\alpha 6$ antibody, the recombinant human integrin $\alpha 6\beta 1$ used in our *in situ* integrin overlay assays exhibits very weak affinity for laminin-111 (Kiyozumi et al., *Life Sci. Alliance* 2018;1:e201800064). We also performed the *in situ* integrin overlay assays with recombinant mouse integrin $\alpha 6\beta 1$ (m $\alpha 6\beta 1$), which gave positive signals on the ExE basement membrane. These results have been included in **Supplementary Figure S2** and described in the **Results and Discussion** (page 8, lines 10–13, revised manuscript).

Figure 4: Can the TSCs also bind to laminin 511 and to laminin 332 (as a control)? While the experiments performed with laminin 111 in which the Glu1605, which is required for integrin binding, is replaced with Gln show that integrins are involved, this does not support binding of specifically $\alpha 7\beta 1$ or $\alpha 3\beta 1$, as this residue is also required for $\alpha 6\beta 1$ binding to laminin 111 and 511.

- TSCs were previously shown to adhere to laminin-511 or -521 (Klaffky et al., *Dev. Biol.* 2001;239:161-175). We have referred to this previous work in the **Results and Discussion** (page 9, lines 7–8, revised manuscript).

Are the differences in proliferation of TSCs on laminin 111 compared to uncoated dishes statistically significant? The authors explain the high proliferation on uncoated plastic as

binding of the cells to fibronectin or vitronectin for serum from the cell culture medium - can this binding be blocked for example with RGD, anti-integrin $\alpha 5$ or anti-integrin $\beta 1$ Ab? Inclusion of such controls on both the cells plated on uncoated and laminin 111 coated dishes would strengthen the data.

- There was a significant difference ($P < 0.05$) for cell proliferation on laminin-111-coated and non-coated dishes, as shown in the revised **Figure 4B**.
- The adhesion of TSCs onto 70CM-coated dishes was blocked by anti- $\beta 1$, anti- $\beta 3$, and anti- αV integrin antibodies, as shown in the newly added **Supplementary Figure S3** (page 10, lines 3–4, revised manuscript).

Labelling of immunofluorescence images in Figure 5 need to be clearer - here too it is not clear what the white staining is in the left panel of B (presumably laminin 1).

- The white color in **Figure 5B** indicates the area double-positive for the anti-laminin $\alpha 1$ antibody (magenta) and recombinant integrin $\alpha 7 \times 2 \beta 1$ (green), because white is the consequence of superimposing magenta and green. We have amended the legend for **Figure 5B** to include this information (page 29, lines 15–17, revised manuscript).

Reviewer #2

In this report, the authors investigate a role for laminin in the trophoblast stem cell (TSC) niche. The authors utilize a variety of research approaches to examine laminin-integrin signaling in the early mouse embryo. The research provides an important scientific advance. Some concerns with the experimental design and interpretation of results are presented below.

1. Page 5, the comment that Cdx2 is indispensable for establishment of TSCs is possibly an overreach. Reprogramming experimental results published in Cell Stem Cell in 2015 demonstrated that Cdx2 was not needed in the transcription factor cocktail to generate TSCs.

- We have amended the text to indicate that CDX2 is dispensable for transdifferentiation of TSCs (Kubaczka et al. *Cell Stem Cell* 2015;17:557-568) in the **Introduction** (page 5, lines 4–6, revised manuscript).

2. The integrin binding assay requires some additional details, especially the method of detecting integrin binding. This was apparently accomplished with the anti-Velcro antibody. What is "Velcro"? What does the anti-Velcro antibody recognize? What is the source of the anti-velcro antibody?

- We apologize for the confusing description in the original manuscript. The recombinant integrins used were truncated soluble forms lacking the transmembrane and cytoplasmic domains of both the α and β subunits. To secure their dimerization, two peptide segments rich in acidic residues (termed "ACID") or basic residues (termed "BASE") are fused to the C-termini of the α and β subunit extracellular regions, respectively (Takagi et al., *Nat. Struct. Biol.* 2001;8:412-416). Because the ACID and BASE peptides interact with one another like "Velcro" (O'Shea et al., *Curr. Biol.* 1993;3:658-667) (see revised **Figure 2A, center**), the ACID-BASE complex is named "Velcro". The anti-Velcro antibody specifically recognizes recombinant integrins (see revised **Figure 2A, right**). The anti-Velcro antibody was raised in rabbits by immunization with ACID and BASE peptides as immunogens (Sato-Nishiuchi et al, *J. Biol. Chem.* 2012;287:25615-25630).

3. *The authors provide information about laminin promoting adhesion and proliferation of TSCs. Do the TSCs cultured on laminin retain their abilities to differentiate into trophoblast giant cells, and other differentiated trophoblast lineages? Some evidence about the capacity for trophoblast differentiation would be helpful.*

- It remains to be addressed whether TSCs cultured on laminin retain their differentiation potential. However, TSCs can be established from blastocysts under defined conditions by culture on Matrigel (Kubaczka et al., *Stem Cell Reports* 2014;2:232-242). Because laminin is the major component of Matrigel, TSCs cultured on laminin may well retain their differentiation potential. We have added this information to the **Results and Discussion** (page 10, lines 16–20, revised manuscript).

4. *In the literature there is some evidence for the importance of IV collagen as a contributor to the TSC niche. See the following reports:*

Schenke-Layland K, Angelis E, Rhodes KE, Heydarkhan-Hagvall S, Mikkola HK, Maclellan WR. Collagen IV induces trophoectoderm differentiation of mouse embryonic stem cells. Stem Cells. 2007 Jun;25(6):1529-38. Epub 2007 Mar 15. PubMed PMID: 17363553.

Okae H, Toh H, Sato T, Hiura H, Takahashi S, Shirane K, Kabayama Y, Suyama M, Sasaki H, Arima T. Derivation of Human Trophoblast Stem Cells. Cell Stem Cell. 2018 Jan 4;22(1):50-63.e6. doi: 10.1016/j.stem.2017.11.004. Epub 2017 Dec 14. PubMed PMID: 29249463.

Some comments may be helpful.

- In humans, TSCs can be established from blastocysts cultured on collagen type IV *in vitro* (Okae et al., *Cell Stem Cell* 2017;22:50-63). It was also reported that collagen type IV can induce trophoectoderm differentiation of mouse embryonic stem cells *in vitro* (Schenke-Layland et al., *Stem Cells* 2007;25:1529-1538). Because TSCs do not express collagen-binding integrins (**Figure 1A**), collagens may contribute to the ECM niche for TSCs in an integrin-independent manner. We have included this additional discussion in the **Results and Discussion** (page 12, lines 4–9, revised manuscript).

5. *The culture medium used in the in vitro analysis is important, especially the presence of FBS. Information about the generation of the MEF conditioned culture medium should be provided. Was FBS in the MEF culture medium?*

- MEF-CM was prepared by culturing mitomycin C-treated MEFs in TS medium, which includes 20% FBS (Hayakawa et al., *Curr. Protoc. Stem Cell Biol.* 2015;32:1E.4.1-32). We have added this information to the **Materials and Methods** (page 16, lines 18–19, revised manuscript).

6. *Fig. 1: The integrin profile in cultured TSCs is useful. What is the integrin profile of a trophoblast stem cell developing in situ? What is the basis of the assumption that the culture conditions for TSCs fully replicate the in vivo environment?*

- It is difficult to specifically identify TSCs *in vivo*. However, the trophoectoderm cells in the blastocyst were reported to express the integrin $\alpha 6$ and $\alpha 7$ subunits (Klaffky et al., *Dev. Biol.* 2001;239:161-175). TSCs cultured *in vitro* were shown to contribute to the placenta *in vivo* when implanted into blastocysts (Tanaka et al., *Science* 1998;282:2072-2075), offering a basis for the assumption that the *in vitro* culture conditions for TSCs replicate the *in vivo* environment.

7. Fig. 2: It is not clear how the integrin binding was detected.

- The recombinant integrins used were truncated soluble forms lacking the transmembrane and cytoplasmic domains of both the α and β subunits. To secure their dimerization, peptide segments rich in acidic residues (termed “ACID”) or basic residues (termed “BASE”) are fused to the C-termini of the α and β subunit extracellular regions, respectively (Takagi et al., *Nat. Struct. Biol.* 2001;8:412-416). Because the ACID and BASE peptides interact with each other like “Velcro” (O’Shea et al., *Curr. Biol.* 1993;3:658-667), the ACID-BASE complex is called “Velcro”. The recombinant integrins bound to the tissue sections were specifically detected by the anti-Velcro antibody. Please refer to the revised **Figure 2A**.

8. Fig. 4: Do the ECM proteins used in the cell adhesion or proliferation assays contain contaminating cytokines or growth factors? What is the basis for determining purity of the ECM protein preparations? If contaminants are present, then could they influence TSC proliferation?

- Recombinant ECM proteins were purified by His-tag affinity purification. Because the same affinity purification was employed for both recombinant laminin-111 and its inactive mutant (laminin-111 EQ), any potential contaminants would be equally contained in the wild-type and mutant proteins.

9. Fig. 4: In panel C, rLM-111 +70CMF4H should be included as a positive control to compare with the rLM-111 -70CMF4H.

- The experiments shown in **Figure 4C** were intended to determine whether laminin-111 can substitute for the functions of FGF4 and MEF-CM. We believe that non-coated +70CM+F4H is sufficient as a positive control.

10. Fig. 5, panel C: What is the specificity of the Cdx2 antibody? It looks there is extensive Cdx2 immunoreactivity throughout the adjacent uterine decidua.

- Because the anti-CDX2 mAb is a mouse IgG, fluorescence-labeled anti-mouse IgG was used as a secondary antibody. Therefore, the uterine decidua was non-specifically stained by the secondary antibody because the maternal tissue contained endogenous mouse IgG.

Reviewer #3 (Comments to the Authors (Required)):

The authors have examined the role of extracellular matrix in trophoblast stem cell (TSC) biology. They have concluded that laminin-111 interaction with integrin $\alpha7\beta1$ plays a critical role in the adhesion, growth and maintenance of these cells based on both in vitro and in vivo experiments.

Overall the study has been carefully carried out with the major findings supported by the data. By determining the integrin and laminin subunits expressed at the trophoblast basement membrane (BM) zone and the integrins that bind to the BM, the authors were able to deduce the major interaction that occur in situ. In addition, the authors show that laminin-111 supports TSC adhesion and proliferation in the presence of F4H and heparin. Particularly persuasive is the comparison made between wild-type and laminin- $\gamma1$ E:Q mutant mouse embryos, the latter unable to bind to integrins, in which it was found that the population of TSC cells is reduced in the absence of the specific integrin binding to the laminin.

Specific questions (minor concerns):

1. Fig. 2. The immunofluorescence colors are magenta (integrin) and green (laminin). In the methods, Alexa 488 (green) is listed as the reagent used to detect applied integrin via the anti-Velcro antibody while Alexa 546 (yellow/orange) is used to detect the laminin. This is a bit confusing. Have the authors used pseudo color assignments in the figure as shown?

- We apologize for the confusing description in the original manuscript. The colors for integrin and laminin were changed by the software in **Figure 2**.

2. Fig. 2. It would help the reader to see larger images of the region of the extraembryonic BM.

- In accordance with the reviewer's suggestion, we have added magnified images of the region in the extraembryonic basement membrane wherein the integrins yielded positive signals in the revised **Figure 2B**.

3. page 8, upper paragraph. Integrin $\alpha 7 \times 2 \beta 1$. Is there a phenotype in the integrin $\alpha 7$ knockout mouse known to correspond to an alteration of trophoblast stem cells?

- Placental defects were reported in integrin $\alpha 7$ knockout mice (Welser et al., *Placenta* 2007;28:1219-1228). However, it remains uncertain whether these defects were the consequence of altered behavior of TSCs.

We would like to thank the reviewers for their insightful comments, which have helped us to improve the manuscript with additional data. We hope that the revised manuscript will be considered acceptable for publication in *Life Science Alliance*.

December 31, 2019

RE: Life Science Alliance Manuscript #LSA-2019-00515-TR

Prof. Kiyotoshi Sekiguchi
Osaka University
Institute for Protein Research
3-2 Yamadaoka
Suita, Osaka 565-0871
Japan

Dear Dr. Sekiguchi,

Thank you for submitting your revised manuscript entitled "LAMININ IS THE ECM NICHE FOR TROPHOBLAST STEM CELLS". As you will see, the reviewers appreciate the changes introduced in revision, and we would thus be happy to publish your paper in Life Science Alliance pending final minor revisions:

- Please address the remaining concerns of reviewer #1 by changes to the text
- Please make sure to indicate the technical/biological replicates for all data (eg., currently unclear for RT-PCRs)
- Please upload all supplementary figures as individual files and without figure legends, these should get included in the main manuscript file
- Please provide Table S1 in either excel or word docx format

A. FINAL FILES:

-- Summary blurb (enter in submission system): A short text summarizing in a single sentence the study (max. 200 characters including spaces). This text is used in conjunction with the titles of papers, hence should be informative and complementary to the title. It should describe the context and significance of the findings for a general readership; it should be written in the present tense and refer to the work in the third

person. Author names should not be mentioned.

B. MANUSCRIPT ORGANIZATION AND FORMATTING:

Sincerely,

Reviewer #1 (Comments to the Authors (Required)):

The manuscript has improved with the inclusion of details concerning experiments and explanations of staining patterns observed, which have been largely included into the methods section. In several cases, it would facilitate reading of the manuscript if this formation was also be included into the figure legends eg Fig 2 B - details concerning whether the laminin staining refers to laminin alpha 1 or laminin gamma1; Fig 4

explanation of 70CM+F4H.

References for the laminin antibodies employed would also strengthen the data as several anti-mouse antibodies have been produced that are not well characterised. Is the anti-integrin antibody used GoH3? Inclusion of such information on antibody/clones names would strengthen the manuscript.

In some sections the text is difficult to follow and would benefit from more simple text.

The assumption from the work presented is that integrin $\alpha 7\beta 1$ mediates binding to laminin 111 (and potentially laminin 511) in the extra-embryonic BM and thereby contributes to implantation. The authors should comment on the integrin $\alpha 7$ knockout mouse which does not have an implantation phenotype, presumably due to the involvement of other non-integrin receptors eg dystroglycan the absence of which indeed results in an implantation phenotype. This would more correctly convey the relative importance of integrins and non-integrin receptors to different steps in the implantation process.

Reviewer #2 (Comments to the Authors (Required)):

The authors have satisfactorily addressed my concerns.

We are happy to hear that the changes made on our manuscript LSA-2019-00515-TR have been appreciated by the reviewers and the revised manuscript will be acceptable pending minor revision.

Here we amended the manuscript as follows:

- 1) The replicates for the RT-PCRs were indicated in the legends for Figure 1A (page 28, line 3).
- 2) The legends for Figure 2B were rephrased to include the information regarding the anti-laminin antibodies used (page 28, lines 13-16).
- 3) 70CM+F4H in Figure 4 has been explained in Non-standard abbreviations (page 2, lines 1-2).
- 4) The information on the anti-integrin antibody on page 13, line 18 has been corrected in the revised manuscript. The anti-integrin antibody used in Figure S3 was anti- α V mAb (RMV-7) but not anti- α 6 mAb (GoH3) (please also refer to page 10, lines 3-4). We apologize for the confusing description on page 13.
- 5) As explained in the previous responses to Reviewer #3, integrin α 7 knockout mice exhibit placental defects with partial embryonic lethality (Welser JV et al. *Placenta* 2007; 28:1219-1228). However, it remains uncertain whether the defects were the consequence of altered behavior of TSCs. In this study, we employed integrin overlay assays to probe the integrin ligands in the embryonic basement membranes focusing on those underlying TSCs. We need to gather more data to discuss the involvement of integrin α 7x2 β 1 and other non-integrin receptors, e.g. dystroglycan, in implantation. We would like to address the issue in a separate paper.

We uploaded all supplementary figures as individual files without figure legends. We also uploaded Table S1 in a Word format.

We hope that the revised manuscript is now acceptable for publication in Life Science Alliance.

RE: Life Science Alliance Manuscript #LSA-2019-00515-TRR

Prof. Kiyotoshi Sekiguchi
Osaka University
Institute for Protein Research
3-2 Yamadaoka
Suita, Osaka 565-0871
Japan

Dear Dr. Sekiguchi,

Thank you for submitting your Research Article entitled "LAMININ IS THE ECM NICHE FOR TROPHOBLAST STEM CELLS". It is a pleasure to let you know that your manuscript is now accepted for publication in Life Science Alliance. Congratulations on this interesting work.

DISTRIBUTION OF MATERIALS:

Again, congratulations on a very nice paper. I hope you found the review process to be constructive and are pleased with how the manuscript was handled editorially. We look forward to future exciting submissions from your lab.

Sincerely,

Andrea Leibfried, PhD
Executive Editor
Life Science Alliance
Meyerhofstr. 1

69117 Heidelberg, Germany
t +49 6221 8891 502
e a.leibfried@life-science-alliance.org
www.life-science-alliance.org